# Decompose Boolean Matrices with Correlation Clustering

**DOI:** 10.3390/e23070852

**Published:** 2021-07-02

**Authors:** László Aszalós

**Affiliations:** Faculty of Informatics, University of Debrecen, Egyetem tér 1, 4032 Debrecen, Hungary; aszalos.laszlo@inf.unideb.hu

**Keywords:** matrix decomposition, correlation clustering, similarity

## Abstract

One of the tasks of data science is the decomposition of large matrices in order to understand their structures. A special case of this is when we decompose relations, i.e., logical matrices. In this paper, we present a method based on the similarity of rows and columns, which uses correlation clustering to cluster the rows and columns of the matrix, facilitating the visualization of the relation by rearranging the rows and columns. In this article, we compare our method with Gunther Schmidt’s problems and solutions. Our method produces the original solutions by selecting its parameters from a small set. However, with other parameters, it provides solutions with even lower entropy.

## 1. Introduction

Decomposition of matrices is used in many areas. Decomposition or factorization typically simplifies the solution of specific problems. For example, consider the LU-decomposition when, for a square matrix *A*, we need to determine a lower triangular matrix *L* and an upper triangular matrix *U* such that A=LU. This can be used to solve linear equations numerically or to calculate the determinant. A partially similar method is the *Boolean matrix factorization* [1], where Boolean algebra operations should be used instead of matrix multiplication. The goal in this case is the same: to make two factor matrices simpler than the original. A special case of this problem is when the matrix is symmetric and the two factor matrices are each other’s transposes [2].

However, there is also a variant of the decomposition of matrices where we do not want to convert the matrix into a product, but break it down into independent, often homogeneous blocks by rearranging the rows and columns [3]. Since we want to group both the rows and columns of the matrix—these processes are not independent of each other, and we finally cut the matrix into blocks—this task has been given several names, e.g., biclustering, block clustering, etc. The entropy of most of the blocks of the solution is 0, as they contain constant values. However, in addition to clustering, the task can be solved in many other ways, see [4,5,6]. A natural generalization of this decomposition task is when the matrix contains not binary but real values; then, the decomposition gives a nice visualization of the clusters, as at the solution the sample variance of the blocks of the decomposition is minimal. Biclustering has many applications in bioinformatics; Ref. [7] gives a detailed list of biclustering methods. Mostly, we have heterogeneous relations or measurements, e.g., in a recommendation system the rows and columns denote products and customers, and we want to group both of them to target the customers more precisely. For biclustering we need to cluster a bipartite graph, while the standard clustering methods cluster a normal graph by transforming the distance/similarity of objects into a value and produce clusters based on this.

In this article, we only examine the case of logical matrices of relations, i.e., the cases when the matrix can contain only 0 s and 1 s. In some practical tasks—e.g., at recommendation systems—we expect such a rearrangement where the non-diagonal blocks in the resulting matrix contain only 0 s. Gunther Schmidt wrote in detail about the significance of these tasks and presented methods for this rearrangement in [8]. In this article, we will use his examples and compare his solutions with those generated by our software.

Recently, the study of Ferrer-type relations and their nestedness property has received more attention [9,10], which are well handled by the method we propose. Moreover, the Boolean matrix factorization is a popular method for exploring the structure of Boolean matrices, but as it is an approximation method, we lose information with it. Similar to Schmidt’s methods, we partitioned the rows and columns of the matrix—these are lossless methods—but we can refine the decomposition by choosing the parameters of the method.

The structure of the article is as follows: in Section 2, we present the theoretical background necessary to understand the article. Here, we will talk about the concept of similarity, the coefficients defined for similarity, rough set theory, the coefficient variant we obtain with upper approximation and finally correlation clustering. Section 3 presents the method of decomposition of heterogeneous relations and the results obtained in this way. In Section 4, we explain why the method of the previous section cannot be used for homogeneous relations, and what needs to be modified to make this method work in this case as well. Furthermore, we present a few more relations and the results obtained for them, which nuances the properties of our method. Finally, we summarize the results of the article.

## 2. Theoretical Background

We want to use clustering to group rows and columns of a matrix. The principle of clustering is usually to group *close* objects into a common group and *distant* objects into separate groups. If we cannot describe our objects with *n* real numbers, but with categorical features, then the use of distance is cumbersome. For example, how far apart are two books, two persons, or two rows of a matrix of a relation? In such cases, it is worth using the concept of similarity. Therefore, we replace the concept of distance with the concept of similarity at clustering. Correlation clustering is unique among clustering procedures in that it is based on similarity. As this is not a well known method, we will briefly present it here. We are using the upper approximation to calculate similarity of objects, which is a concept of rough set theory, so first we describe the main points of rough set theory.

### 2.1. Rough Set Theory

In case of traditional set theory, there are two possibilities: a given element is either included in a given set or not included in it; hence, the membership is a bivalent condition. In fuzzy set theory, the elements have degrees of membership. The degree can be any real number between 0 and 1 (inclusive). If 0<r<1, then the given element belongs only partially to the set, the larger *r* means higher degree of membership. Rough set theory lies between traditional and fuzzy set theory; here, we have three possibilities (i.e., trivalent relation):The element is definitely not in the set;The element must belong to the set;The item may belong to the set.

What is the source of this uncertainty? Consider, e.g., a heterogeneous relation where rows correspond to patients and columns to symptoms. It is possible for two patients to produce exactly the same symptoms, but only one of them is affected by a given disease. However, by only using the symptoms we cannot decide which one is sick; they are indistinguishable (or indiscernible) based on these criteria [11].

Indistinguishability is, of course, an equivalence relation, and instead of all objects we can work with their quotient sets; because we either consider all the elements of a given equivalence class or none of them (as these elements are indistinguishable from each other). Quotient sets are indicated by small squares on the right side of Figure 1, which give a partition of the base set, i.e., the set of objects.

Returning to the previous example, if a symptom is a marker of a particular disease—i.e., if a symptom appears, then the patient is really ill—then that symptom can be used as a *lower approximation* of the set of ill patients because the person being marked is obviously ill. There may also be symptoms that exclude a particular disease—that is, whoever has this symptom is certainly not ill. If we take the contraposition of this statement, the absence of a particular symptom allows a person to be ill, but does not guarantee it. Thus, we can treat the set of people without this symptom as an *upper approximation* of the set of ill patients.

The question is how to handle this information. For example, an expensive treatment may be available only to proven patients (lower approximation), while isolation is mandatory for all who are suspected as having the disease (upper approximation). Let the base set be *U*, the indistinguishability equivalence relation be ϱ, and the set to be approximated be *A*. Then, the upper and lower approximations are defined as follows:$$\bar{appr}\left(A\right)=⋃\left\{X\in U/\varrho |A\cap X\neq \emptyset \right\}\mathrm{and}\;\underline{appr}\left(A\right)=⋃\left\{X\in U/\varrho |X\subset A\right\}.$$

### 2.2. Similarity

If the concept of distance cannot be used, we assign a real number to any pair of objects based on the identity or difference of the characteristics describing the objects. For complete match, this value is usually 1, while for complete difference it is 0. The value is determined on the basis of the number of common and different characteristics.

There are several calculation methods for similarity, of which the Jaccard coefficient is the simplest and maybe the most common:(1)$$J\left(A,B\right)=\frac{\left|A\cap B\right|}{\left|A\cup B\right|},$$
i.e., this calculates the proportion of common features among all features (Figure 1 on left). In the article, we will use a variant of this; instead of the original sets, we will work with their upper approximation:(2)$$J^{\prime }\left(A,B\right)=\frac{|\bar{appr}\left(A\right)\cap \bar{appr}\left(B\right)|}{|\bar{appr}\left(A\right)\cup \bar{appr}\left(B\right)|}.$$

For the intersection and the union, we consider the quotient sets whose intersection with sets *A* and *B*—i.e., with the little squares in Figure 1 on the right—is not empty. In case of an intersection, both sets must each contain at least one (possibly different) object of the quotient set, while in case of a union, it is sufficient if the given square contains an object of one of the sets. Treat this definition with caution because the upper approximation of the intersection is not equal with the intersection of upper approximations.

Consider Figure 2, which contains a binary matrix, so it can only contain 0 s and 1 s. Let us assume that the columns of the matrix are clustered, where only three clusters are interesting for us: $C_{k}$, $C_{l}$ and $C_{n}$, as the examined rows *i* and *j* only contain 1 s in the columns of these clusters. If one cluster in these two rows contains at least one 1, we mark all the 0 s of the cluster in the containing row with ∗. We do this, as we think that these 0 s could be 1 s. Considering that there are two readers who love the same books, a clustering should put them in the same cluster. If one of them buys a new book, then we can expect that the other will also buy the same book. This option is indicated by the ∗s.

If we take the original problem—to decompose a binary matrix of relation *R*—then we are interested in its minimal difunctional relation h(R) [8]. A relation *R* is *difunctional* if $R;R^{T};R=R$ holds, where the semicolon is the operator of the composition. Moreover, h(R)=inf{H|R⊆H and H difunctional}, i.e., the least difunctional relation which contains the relation *R*. The matrix decomposition problem can also be formulated by determining the relation h(R) for a given relation *R*. However, we cannot calculate this efficiently according to this definition; therefore, we chose another method. Roughly speaking, we obtain relation h(R) from relation *R* by *rewriting* the 0 s to 1 s in the blocks containing 1 s, so these 0 s could be 1 s. In Figure 2, there are six 1 s in 5 columns in the two marked rows. Hence, the original Jaccard coefficient is J(i,j)=1/5 as we have one common column out of five. On the other hand, we have 11 columns containing 1 s or ∗s in rows *i* and *j*, but only 3 columns contain them in both rows. Therefore, the modified coefficient is $J^{\prime }\left(i,j\right)=3/11$.

How do we interpret it when the value of the modified coefficient is *x*? The Pearson’s correlation coefficient may be familiar from statistics. This can take a value ranging from −1 to 1; if there is a direct linear dependence between the two random variables examined, then this value will be 1, and in case of an inverse linear dependence this will be −1. Intermediate values indicate a partial dependence. For correlation clustering, any pair of objects has a value from an interval [−1,1]. These can be considered as attraction and repulsion between objects: negative values mean repulsion, positive values mean attraction. A higher (absolute) value means the greater the force of attraction/repulsion. During clustering, we want to arrange similar objects into a common cluster (these objects should attract each other), while different objects should be in different clusters (these should repulse each other). Similarities and differences of rows and columns should be transformed into attraction and repulsion. It may be clear that similarity means attraction, difference repulsion, but how can this be quantified? Where is the point on the similarity–difference scale that will mean neural attraction–repulsion? By definition, our Jaccard coefficients (and their variants) can be taken from the interval [0,1], while the repulsion–attraction is interpreted on the interval [−1,1]—that is, we want to use monotone functions t:[0,1]→[−1,1] that are smooth enough. Figure 3 contains two such functions—there are an infinite number of other functions that satisfy these conditions—and the subject determines which one to use. To co-write an article can be considered a commitment between two people. Being co-authors means they should not be considered different, but this may not be enough for a minimum degree of similarity between them. In comparison, travelling on the same train does not make two people similar. It is a coincidence, people may still have completely different styles. Therefore, we feel it is worth leaving the user to select the right conversion function for the subject. We used the black conversion function ($t\left(x\right)=2\sqrt[{5}]{x}-1$) in Figure 3 for all relations in this article with one exception: the Ferrer-type ([8] p. 26) relation presented later needs the red conversion function ($t\left(x\right)=2x^{3}-1$). The matrix of this relation could be considered dense, and we accept similarity only for high coefficients. The other matrices are mostly sparse, and yet a low coefficient means similarity.

### 2.3. Correlation Clustering

Zahn [12] asked *which equivalence relation is closest to a given tolerance relation?* The distance between the tolerance (reflexive and symmetric) relation *R* and the equivalence relation *E* is the value |EΔR|, where *Δ* denotes the symmetric difference. This problem is the correlation clustering. Since the equivalence relation *E* can be considered a partition of the base set (the set of objects), and the number of possible partitions is given by the Bell numbers (which grow very fast), in practice (except for a few special cases), only an approximate solution can be given if we have more than 15 objects.

There are many ways to find a nearby equivalence relation for a given relation *R*, many of which have been implemented in [13]. In practice, we used a method that was developed by us. Here, we treat objects as particles with electric charge, and object attraction or repulsion is fulfilled according to their charge. Unlike electricity, here similar charges—more precisely, similar objects—are attracted to each other, and different ones repel each other. Let $f_{xy}$ be the attraction for every object pair *x*, *y*. This attraction defined between objects can be extended by superposition to object-to-cluster and cluster-to-cluster attraction:$$f_{xh}=\sum_{y\in h}f_{xy},\mathrm{and}f_{gh}=\sum_{x\in g}f_{xh}.$$

We can arrange forces $f_{xy}$ into a matrix *F*, where $F_{x}$ denotes the column vector of object *x* in *F*. Similarly, we can represent a clustering as a matrix *C*: let $c_{gx}$ be 1 if object *x* is a member of cluster *g*.

Our method starts with singletons: each object—in this case, row or column—represents a one-element cluster—i.e., matrix C at the beginning is diagonal. Then, we always move object *x* to the most attractive cluster *h*, for which $f_{xh}$ is maximal (step move). For this, we need to calculate the matrix product $CF_{x}$ for each object *x*. The biggest numbers in this vector give the most attractive clusters. A special case of this is when an object is repelled by every cluster, even its own—i.e., all numbers of $CF_{x}$ are negative. Then, we create a new cluster that contains only that one object. As an object does not attract or repulse itself, this is an ideal situation for it. Once all the items are in their most attractive cluster, we can move on.

If there are attractive clusters—i.e., force $f_{gh}$ is positive for two different clusters *g* and *h*, i.e., matrix product $CFC^{T}$ has some positive, non-diagonal element—then we merge the most attractive clusters *g* and *h*, where the force $f_{gh}$ is maximal (step join). If we have reached an equilibrium state—where no object wants to move, and there are no attracting clusters—we cannot reduce the distance between the *R* relation and the partition generated by the clusters, and then we stop and the current clusters give the equivalence classes. In Figure 4, the coloured background indicates the correlation clusterings.

Due to the matrix multiplications, we might think that the complexity of the steps move and join is cubic, but the matrix *C* is sparse, so the complexity of these steps is closer to quadratic. During step calc, the symmetric square matrix *F* is calculated, so the complexity here is $O(r^{2}c)$ or $O(rc^{2})$. However, we cannot predict in advance how many cycles will be executed, but the method converges (the distance mentioned before is decreasing), and so eventually halts.

## 3. Decomposing Heterogeneous Relations

In the previous subsection, we did not specify how to determine the value of forces $f_{xy}$. In case of heterogeneous matrices, the rows and columns are not clustered simultaneously, but this does not mean independent clustering. To cluster rows, we need to calculate the similarity of the rows (step calc). We have already seen that for this we need the result of the clustering of columns and we need to calculate the coefficients for any two rows based on (Equation 2). Here, the upper approximation of a row is the sum of the sizes of the column clusters which contain 1 in this row. Fortunately, it is enough to calculate the similarity of the rows once—just before correlation clustering—and we can use these values of $f_{xy}$ during the clustering, as they do not change. Once the clustering of rows has reached a standstill, the columns may be clustered. To do this, we need the similarity of the columns, which can be calculated in a similar way, i.e., by using the clustering of rows. Once we are done with that, we can go back to the clustering of rows. If neither the clusters of rows nor the clusters of the columns change, we are finished.

Figure 5. contains the relation ([8] p. 22). We started with singleton clusters of rows and columns, and we obtain these clusters as the results of the consecutive correlation clustering:
rows:{1,11,9,17},{16,15},{3,6,7},{2},{4,13},{14},{8,10,12,5}columns:{3,13,6,7},{8,1,4},{9,10,5},{2},{11,12}rows:{1,11,9,15,16,17},{2,3,6,7},{4,13,14},{8,10,12,5}columns:{3,13,6,7},{8,1,4},{9,2,10,5},{11,12}

This gives us the original solution so we are ready but the algorithm still does a cycle to check if there are any further modifications.

The bottom 4 rows and the last 2 columns are empty in this matrix. Our definition would give 0/0 as a coefficient for such empty rows or columns. However, we interpreted this to be 1, given that two blank rows are basically the same. If, on the other hand, we interpret this coefficient as 0, then this means that empty rows are different from everything—even from themselves—and then these 4 rows form 4 separate clusters, and, similarly, the 2 columns do so as well.

If we calculate the *information content* [14] of the relation in Figure 5 without and with clustering, by using
$$I(\frac{x}{x+y},\frac{y}{x+y})=-\frac{x}{x+y}\log_{2}\frac{x}{x+y}-\frac{y}{x+y}\log_{2}\frac{y}{x+y}$$
for the whole matrix and also for its submatrices (weighted by their size), then we obtain around 0.468 and 0.214, respectively, so, by clustering, we obtain a more *organized relation*. Here, *x* denotes the 1 s in the block, while *y* denotes the number of 0 s, so the *p* in the binary entropy function H(p) can be given by x/(x+y). With three exceptions, we obtained homogeneous blocks for which H(0)=0. For the three exceptions, H(10/24)=H(7/10)≈0.80 and H(6/12)=1, and then these values are taken into account with weights 24/221, 12/221 and 12/221 to give 0.214 for entropy after clustering.

Figure 6 contains the only relation in [8], where the black conversion function in Figure 3 was not suitable. However, here the red function successfully reproduced the the original solution.

## 4. Decomposition of Homogeneous Relations

Heterogeneous relations can be thought of as a bipartite graph, and this is the reason to define the similarity of rows by the identity/difference of columns associated with it. A homogeneous relation can be considered as the subset of a Cartesian product of a set with itself, so the rows will no longer be independent of the columns. As we rearrange the rows, the columns are automatically rearranged in the same way. This is also the reason why we used directed graphs to represent homogeneous relations. Since it is no longer necessary to cluster rows and columns separately, the whole process is simplified—see the right side of Figure 4. It may stand out at first glance that the rectangle of step calc is missing. It is easy to see that rearranging the rows of a heterogeneous matrix does not modify the columns (more precisely their clusters) and vice versa. Due to this, the similarity of rows/columns can be calculated in advance and used over and over again when running steps move and join.

The situation is different for a homogeneous matrix. If you move a row to another cluster of rows, then the clusters of the columns also change because there are no separated clusters of rows and columns. Therefore, it is not possible to perform pre-calculations, and so we need to calculate similarity of rows/columns according to the actual situation. Since the rows/columns are not separated, two phases are not required.

Take relation ([8] p. 12) in Figure 7. What about the similarity of rows 7 and 9 at the bottom of the figure? Our ill-considered answer is based on the previous section: they are similar as they only contain 0 s. However, we did not take into account that we have corresponding columns for these rows, and column 7 contains a 1, but column 9 does not. Therefore, they are different, as the solution clearly shows, as they are in separate clusters.

Now, let us look at two rows that are similar to each other according to the solution—rows 13 and 15 in the top left corner. We can characterize row 13 based on the directed graph of the relation—that is, it can be coded as {o15,i15}, since an edge goes to vertex 15 (outgoing) and one comes from there (incoming). A similar representation of row 15 could be {o13,i13}. If we use the coefficient of (Equation 1), we obtain 0/4, meaning these two rows are different. At the beginning of biclustering—when we have just singletons—the coefficient (Equation 2) gives the same result, which prevents us from achieving the expected decomposition. Therefore, we need a new coefficient for the homogeneous case.

If we only consider where the incoming edges come from, and where the outgoing edges point to, we lose information. Therefore, we can define new coefficient variants where we add the current vertex to the adjacent vertices.
(3)$$J^{\prime \prime }\left(x,y\right)=\frac{|\bar{appr}\left(x\cup x.R\right)\cap \bar{appr}\left(y\cup y.R\right)|}{|\bar{appr}\left(x\cup x.R\right)\cup \bar{appr}\left(y\cup y.R\right)|}.$$
(4)$$J^{\prime \prime \prime }\left(x,y\right)=\frac{|\bar{appr}\left(x\cup x.R\right)\cap \bar{appr}\left(y\cup y.R\right)|+|\bar{appr}\left(x\cup R.x\right)\cap \bar{appr}\left(y\cup R.y\right)|}{|\bar{appr}\left(x\cup x.R\right)\cup \bar{appr}\left(y\cup y.R\right)|+|\bar{appr}\left(x\cup R.y\right)\cup \bar{appr}\left(y\cup R.y\right)|}.$$

We define the vertices of outgoing and ingoing edges in the context of relation:x.R={y∈B|xRy}andR.y={x∈A|xRy}.

In (Equation 3), just the outgoing edges are considered, and in (Equation 4), the outgoing and incoming edges are considered. If we now examine the similarity of rows 9 and 19, then as these rows and the corresponding columns contain only 0 s, we could say that they are similar. If we calculate the new coefficients, then x.R=R.x=∅, i.e., there are no outgoing or incoming edges, so $J^{\prime \prime }\left(9,19\right)=\left|\left\{9\right\}\cap \left\{19\right\}|/|\left\{9\right\}\cup \left\{19\right\}\right|=0/2$, and similarly $J^{\prime \prime \prime }\left(9,19\right)=0/4$. Due to this technical modification, the empty rows/columns are clustered into separate (singleton) clusters. Apart from this, our solutions are the same as the original solutions.

Based on Schmidt’s homogeneous problems, we could not differentiate the coefficients $J^{\prime \prime }$ and $J^{\prime \prime \prime }$ because they gave the same clusterings. Therefore, we constructed a relation. Figure 8 shows this and its decomposition based on coefficient $J^{\prime \prime }$. It can be seen in the upper left corner that not only the diagonal blocks contain 1 s, i.e., this result differs significantly from what is expected. Using the $J^{\prime \prime \prime }$ coefficient, only 4 clusters were created: {1,2,3,4,5,6,7}, {9,10,11,13,14}, {12} and {8} instead of the 7 using the previous approach. This clustering solves the problem of the upper left corner: all the blocks containing 1 s are diagonal. The coefficient $J^{\prime \prime }$ was expected to perform worse than $J^{\prime \prime \prime }$ because it uses less information. For additional relations, we obtained the same result if the resulted clusterings were different.

Our last example in Figure 9 shows what we can accomplish with the conversion function *t*. The starting relation was the *non-relative primes*, i.e., xRy if gcd(x,y)>1. If we only cluster the numbers of the rows listed here, the clustering gives clusters {2,4,6,8,10,12,14,16}, {3,9,15}, {5}, {7}, {11}, {13}, i.e., we have to cluster each number according to its smallest prime divisor [15].

When we have only one big block in the relation, the information content is 0.883. By using the black function in Figure 3, with our method we obtain two singleton clusters containing rows 11 and 13, respectively, while all other rows form a big cluster. This gives the expected solution, i.e., all the 1 s are in the diagonal blocks. In this case, the weighted sum of the information content is 0.743, so this solution is not really convincing, as this relation is very far from a difunctional relation. If we use the red transition function in Figure 3, or just a linear one, all the clusters will be singletons.

A good question relates to whether we can have other solutions. If we use the function $t\left(x\right)=2\times x^{\frac{5}{6}}-1$, then we obtain the decomposition shown in Figure 9. It may look strange, but the blocks typically only contain 0 s or 1 s, so in some ways this should also be treated as a good solution; the information content is just 0.032, which is a big improvement over previous values, although we moved away from difunctional relations with this biclustering, but we believe this is an advantage rather than a mistake, and this direction requires further research.

The Python implementation of the methods and some examples are available at https://github.com/aszalosl/decompose_boolean (accessed on 30 June 2020).

## 5. Conclusions

In this article, we present a method for decomposing heterogeneous relations that use correlation clustering and a similarity coefficient based on rough set theory. We have also shown that, with minor modifications, this method can be used to resolve homogeneous relations. When converting between similarity and attraction–repulsion, the user has a large degree of freedom, so it is worthwhile to examine what characteristics of the relation determine the function to be used.

In recommendation systems, it is customary to recommend additional products based on previous purchases of customers who also purchase the given product. In the case of zillions of products, statistical methods do not necessarily help, it makes sense to decompose the customer–product matrix based on similarity of customers and products. However, then all purchases of a customer interested in functional programming, Polish literature and gardening should not be organized into one cluster, as the former matrix decomposition methods do, but each topic should be treated separately. This is possible with the method we proposed.

We have calculated the weighted Shannon entropy for a decomposition without taking into account the storage requirement of the biclusterization. It would be worthwhile to generalize the concept of entropy and to determine the minimum lossless storage allocation for some specific matrices.

It seems to be a natural generalization of Boolean matrices when the values 0 and 1 are replaced by random (noisy) values from a normal distribution, where the mean is constant for each cluster. Unfortunately, the method described here is not applicable to this case, and we are currently researching what can be used from this method.

## Figures and Tables

**Figure 1 entropy-23-00852-f001:**
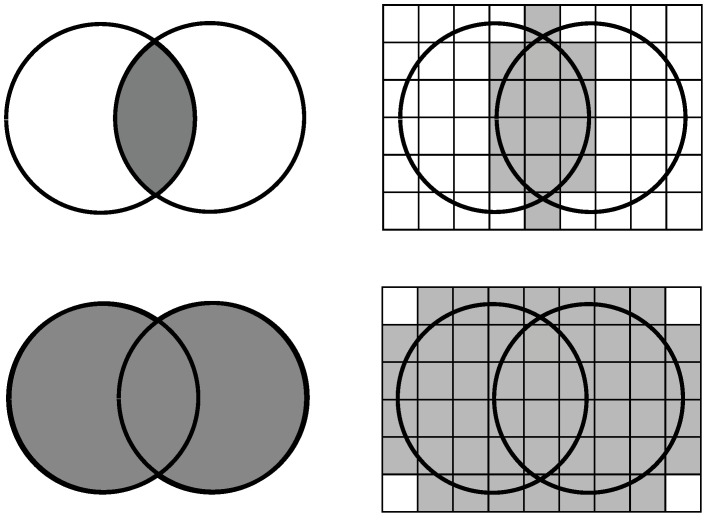
The interpretation of Jaccard’s similarity (**left**) and our suggestion based on rough set theory (**right**).

**Figure 2 entropy-23-00852-f002:**
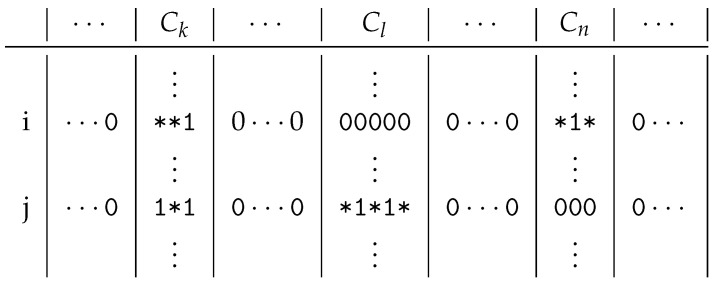
The similarity of two rows of a binary matrix. *J* counts the number of 1–1 pairs in common columns, while $J^{\prime }$ also takes the ∗s into account.

**Figure 3 entropy-23-00852-f003:**
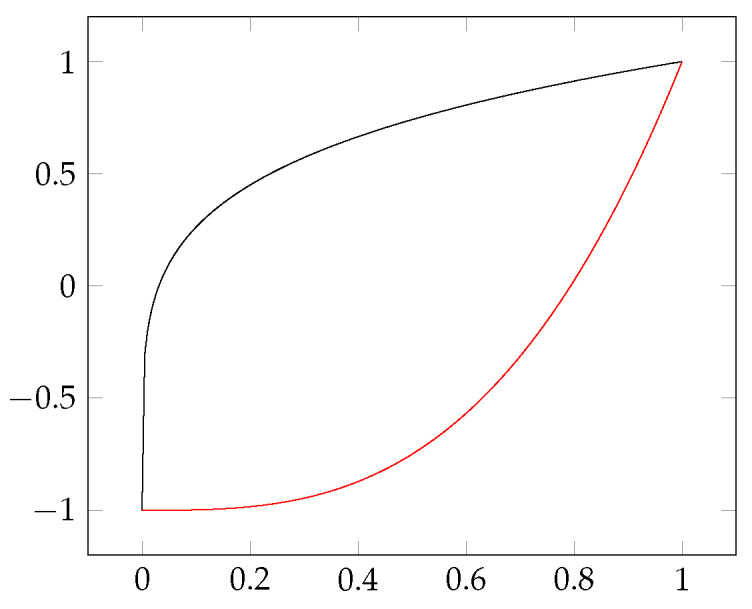
Two functions for conversion between the similarity and the force of attraction: the black is $t\left(x\right)=2\sqrt[{5}]{x}-1$, the red is $t\left(x\right)=2x^{3}-1$.

**Figure 4 entropy-23-00852-f004:**
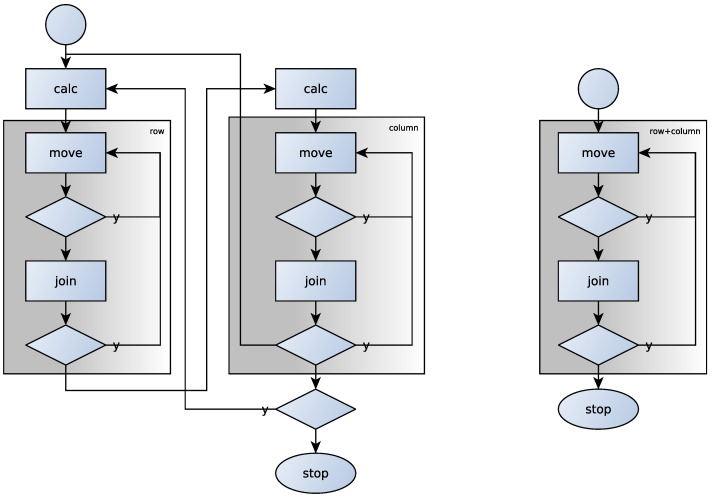
The steps of the correlation clustering in the case of heterogeneous (on the **left**) and homogeneous (on the **right**) matrices.

**Figure 5 entropy-23-00852-f005:**
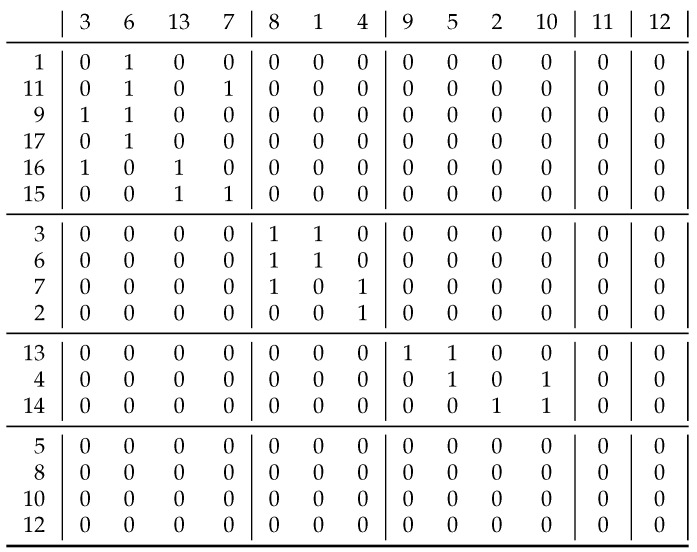
Result of biclustering a sparse heterogeneous relation using the black conversion function in Figure 3.

**Figure 6 entropy-23-00852-f006:**
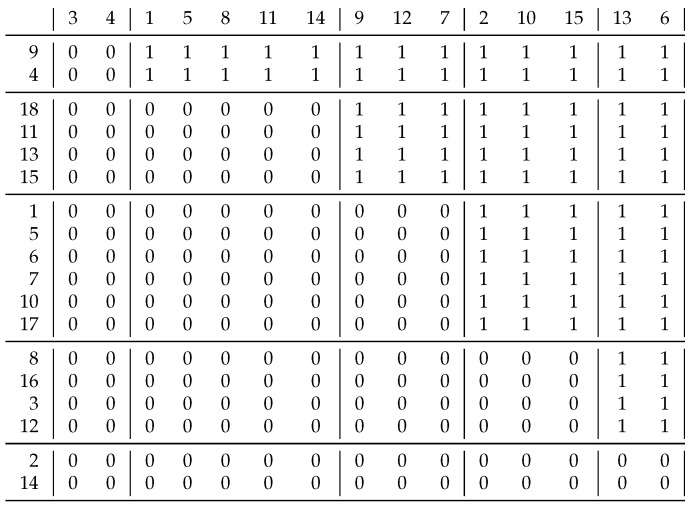
Result of biclustering a Ferrer-type, dense heterogeneous relation, using the red conversion function in Figure 3.

**Figure 7 entropy-23-00852-f007:**
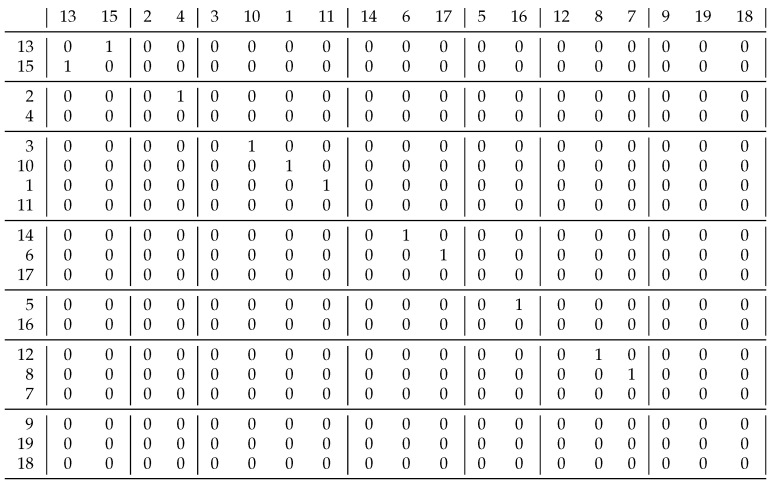
Original decomposition of Schmidt’s homogeneous example. All 1 s are in the diagonal blocks.

**Figure 8 entropy-23-00852-f008:**
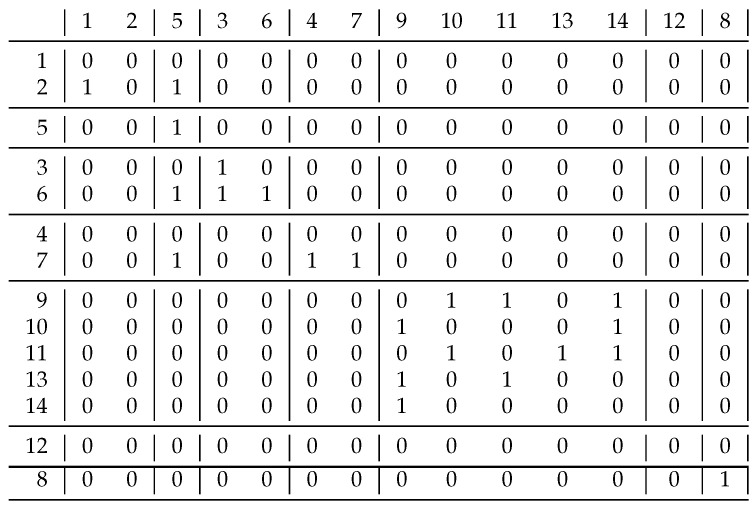
An example relation to differentiate coefficients $J^{\prime \prime }$ and $J^{\prime \prime \prime }$. By using $J^{\prime \prime }$, not only the diagonal blocks contain 1 s.

**Figure 9 entropy-23-00852-f009:**
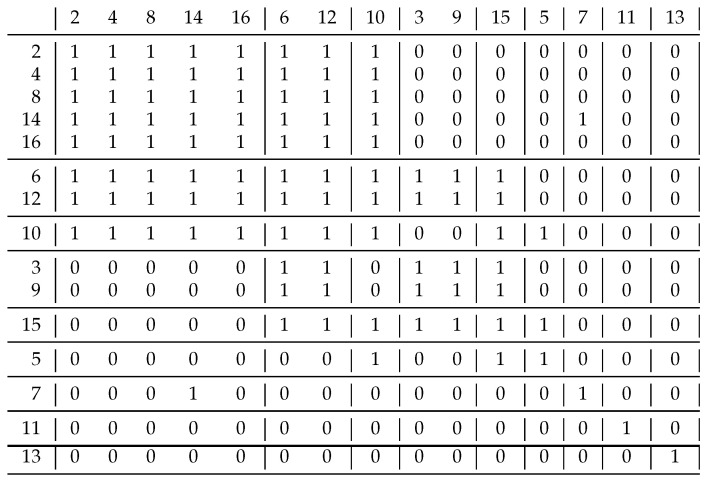
A finer decomposition of the complement of the *relative prime* relation. Not only do the diagonal blocks contain 1 s, but all blocks except two are uniform. If we join any two clusters here, we have more non-uniform blocks.

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
