# Peer review of "Decompose Boolean Matrices with Correlation Clustering"

_entropy, 2021, doi:10.3390/e23070852_

Round 1

Reviewer 1 Report

Dear Sir,

The present manuscript discusses an interesting topic, however, I don't think it is of interest to Entropy. I think it must be submitted elsewhere. From the technical side, the authors discussed a simple idea that may have interesting consequences in set theory in general. If there is a any relation with Entropy, authors must rewrite their article more properly.

Best Regards

Author Response

Dear Sir!

Thanks for reviewing my article.

This article was submitted to a special issue of “Rough Set Theory and Entropy in Information Science”. In this paper, we provide a method for decomposing a Boolean matrix—which can be a customer-product, author-article, etc.—into smaller matrices so that similar products/customers get into the cluster. In the traditional (here homogeneous) case we would cluster a simple graph, whilst in our case it is basically a classification of a bipartite graph (heterogeneous relation). The purpose of the classification is to form matrices that are as homogeneous as possible—i.e. contain just zeros or just ones—thereby reducing the Shannon entropy of the original matrix. The result of calculations was included in the original article, now I have emphasized this during the transformation. According to the theoretical approach, we are looking for the least difunctional relation containing the relation. To find this, our method uses an upper approximation from rough set theory, thus further connecting to the special issue.

Reviewer 2 Report

This paper studies a method for decomposing heterogeneous relations that use correlation clustering and a similarity coefficient based on rough set theory. It is shown that with minor modifications, this method can be used to resolve homogeneous relations. The paper is interesting and the topic is well chosen. I have some comments that may further improve the presentation.

1. The last sentence in the abstract can be better organized. 
2. Line 25, it may be worthwhile noting that clustering here is conceptually similar but practically different from the more commonly known graph clustering. cf. Clustering coefficients of large networks. 
3. The contribution and novelty can be further highlighted in the introduction section. How is the perspective different from previous works?
4. The concepts of upper approximation and lower approximation are interesting. Are they related to some covering problems?
5. Line 121, the definition of h(R) is straightforward. However, it is not clear how it is used in later development.
6. The fact that Jaccard coefficients can be taken from the interval [0, 1], but the repulsion-attraction is interpreted on the interval between -1 and 1 can be better commented.
7. Could you comment the complexity of the algorithm presented in Figure 4?
8. If the clustering of rows has reached a standstill, the columns may be clustered. However, the inverse is not true in general. 
9. It is possible to comment from the perspective of information entropy for the I(.,.) expression.
10. The example in Figure 9 is not easy to observe. I would appreciate some explanation and validation.
11. Is it possible to give some future directions in conclusion? I feel the methodology developed here attractive and potentially applicable in other fields.

Author Response

Thanks for reviewing my article! I rewrote the article according to the comments. However, there are two points I would like to answer here:

4. The concepts of upper approximation and lower approximation are interesting. Are they related to some covering problems? 

In rough set theory, initially the set of the objects was given an equivalence relation and therefore a quotient set (whose elements are called base sets in rough set theory), and then each subset of objects can be approximated with a set of elements of the quotient set. If a given set does not consist of the union of the elements of the quotient set, then two sets could be defined: 1) the union of the elements of the quotient set whose elements are a subset of the given set (this will be the lower approximation), and 2) the union of the elements of the quotient set whose intersection with the original set is not empty (this will be the upper approximation, or a covering of the original set). Later the equivalence relation generalized to coverage and then partial coverage. For practical applications, the (partial) coverage must be finite. So an interesting challenge is how to provide adequate coverage for a specific problem.

8. If the clustering of rows has reached a standstill, the columns may be clustered. However, the inverse is not true in general.

Two correlation clusterings are shown in Figure 4 for the heterogeneous case. When the clustering of the rows (along given column clusters) is completed because it has reached a rest point, we start clustering the columns considering these row clusters. If a rest point is also reached for the columns, the rows are re-clustered with the column clusters obtained in the last step. If neither the clustering of the columns or the rows contained a substantive change, the method ends.

Round 2

Reviewer 1 Report

Authors have improved their work. I recommend its publication.

Reviewer 2 Report

I have no further comment. The paper can be accepted.